# THE LOGICAL EXPRESSIVENESS OF GRAPH NEURAL NETWORKS

**Pablo Barceló**
IMC, PUC & IMFD Chile

**Egor V. Kostylev**
University of Oxford

**Mikaël Monet**
IMFD Chile

**Jorge Pérez**
DCC, UChile & IMFD Chile

**Juan Reutter**
DCC, PUC & IMFD Chile

**Juan-Pablo Silva**
DCC, UChile

## ABSTRACT

The ability of graph neural networks (GNNs) for distinguishing nodes in graphs has been recently characterized in terms of the Weisfeiler-Lehman (WL) test for checking graph isomorphism. This characterization, however, does not settle the issue of which Boolean node classifiers (i.e., functions classifying nodes in graphs as true or false) can be expressed by GNNs. We tackle this problem by focusing on Boolean classifiers expressible as formulas in the logic $FOC_2$, a well-studied fragment of first order logic. $FOC_2$ is tightly related to the WL test, and hence to GNNs. We start by studying a popular class of GNNs, which we call AC-GNNs, in which the features of each node in the graph are updated, in successive layers, only in terms of the features of its neighbors. We show that this class of GNNs is too weak to capture all $FOC_2$ classifiers, and provide a syntactic characterization of the largest subclass of $FOC_2$ classifiers that can be captured by AC-GNNs. This subclass coincides with a logic heavily used by the knowledge representation community. We then look at what needs to be added to AC-GNNs for capturing all $FOC_2$ classifiers. We show that it suffices to add readout functions, which allow to update the features of a node not only in terms of its neighbors, but also in terms of a global attribute vector. We call GNNs of this kind ACR-GNNs. We experimentally validate our findings showing that, on synthetic data conforming to $FOC_2$ formulas, AC-GNNs struggle to fit the training data while ACR-GNNs can generalize even to graphs of sizes not seen during training.

## 1 INTRODUCTION

*Graph neural networks* (*GNNs*) (Merkwirth & Lengauer, 2005; Scarselli et al., 2009) are a class of neural network architectures that has recently become popular for a wide range of applications dealing with structured data, e.g., molecule classification, knowledge graph completion, and Web page ranking (Battaglia et al., 2018; Gilmer et al., 2017; Kipf & Welling, 2017; Schlichtkrull et al., 2018). The main idea behind GNNs is that the connections between neurons are not arbitrary but reflect the structure of the input data. This approach is motivated by convolutional and recurrent neural networks and generalize both of them (Battaglia et al., 2018). Despite the fact that GNNs have recently been proven very efficient in many applications, their theoretical properties are not yet well-understood. In this paper we make a step towards understanding their expressive power by establishing connections between GNNs and well-known logical formalisms. We believe these connections to be conceptually important, as they permit us to understand the inherently procedural behavior of some fragments of GNNs in terms of the more declarative flavor of logical languages.

Two recent papers (Morris et al., 2019; Xu et al., 2019) have started exploring the theoretical properties of GNNs by establishing a close connection between GNNs and the *Weisfeiler-Lehman* (*WL*) test for checking graph isomorphism. The WL test works by constructing a labeling of the nodes of the graph, in an incremental fashion, and then decides whether two graphs are isomorphic by comparing the labeling of each graph. To state the connection between GNNs and this test, consider the simple GNN architecture that updates the feature vector of each graph node by combining it with the aggregation of the feature vectors of its neighbors. We call such GNNs *aggregate-combine GNNs*,

or *AC-GNNs*. The authors of these papers independently observe that the node labeling produced by the WL test always refines the labeling produced by any GNN. More precisely, if two nodes are labeled the same by the algorithm underlying the WL test, then the feature vectors of these nodes produced by any AC-GNN will always be the same. Moreover, there are AC-GNNs that can reproduce the WL labeling, and hence AC-GNNs can be as powerful as the WL test for distinguishing nodes. This does not imply, however, that AC-GNNs can capture every *node classifier*—that is, a function assigning true or false to every node—that is refined by the WL test. In fact, it is not difficult to see that there are many such classifiers that cannot be captured by AC-GNNs; one simple example is a classifier assigning true to every node if and only if the graph has an isolated node. Our work aims to answer the question of what are the node classifiers that can be captured by GNN architectures such as AC-GNNs.

To start answering this question, we propose to focus on *logical classifiers*—that is, on unary formulas expressible in first order predicate logic (FO): such a formula classifies each node $v$ according to whether the formula holds for $v$ or not. This focus gives us an opportunity to link GNNs with declarative and well understood formalisms, and to establish conclusions about GNNs drawing upon the vast amount of work on logic. For example, if one proves that two GNN architectures are captured with two logics, then one can immediately transfer all the knowledge about the relationships between those logics, such as equivalence or incomparability of expressiveness, to the GNN setting.

For AC-GNNs, a meaningful starting point to measure their expressive power is the logic $FOC_2$, the two variable fragment of first order predicate logic extended with counting quantifiers of the form $\exists^{\geq N}\varphi$, which state that there are at least $N$ nodes satisfying formula $\varphi$ (Cai et al., 1992). Indeed, this choice of $FOC_2$ is justified by a classical result due to Cai et al. (1992) establishing a tight connection between $FOC_2$ and WL: two nodes in a graph are classified the same by the WL test if and only if they satisfy exactly the same unary $FOC_2$ formulas. Moreover, the counting capabilities of $FOC_2$ can be mimicked in FO (albeit with more than just two variables), hence $FOC_2$ classifiers are in fact logical classifiers according to our definition.

Given the connection between AC-GNNs and WL on the one hand, and that between WL and $FOC_2$ on the other hand, one may be tempted to think that the expressivity of AC-GNNs coincides with that of $FOC_2$. However, the reality is not as simple, and there are many $FOC_2$ node classifiers (e.g., the trivial one above) that cannot be expressed by AC-GNNs. This leaves us with the following natural questions. First, what is the largest fragment of $FOC_2$ classifiers that can be captured by AC-GNNs? Second, is there an extension of AC-GNNs that allows to express all $FOC_2$ classifiers? In this paper we provide answers to these two questions. The following are our main contributions.

- We characterize exactly the fragment of $FOC_2$ formulas that can be expressed as AC-GNNs. This fragment corresponds to *graded modal logic* (de Rijke, 2000), or, equivalently, to the *description logic* $\mathcal{ALCQ}$, which has received considerable attention in the knowledge representation community (Baader et al., 2003; Baader & Lutz, 2007).

- Next we extend the AC-GNN architecture in a very simple way by allowing global *readouts*, where in each layer we also compute a feature vector for the whole graph and combine it with local aggregations; we call these *aggregate-combine-readout GNNs* (ACR-GNNs). These networks are a special case of the ones proposed by Battaglia et al. (2018) for relational reasoning over graph representations. In this setting, we prove that each $FOC_2$ formula can be captured by an ACR-GNN.

We experimentally validate our findings showing that the theoretical expressiveness of ACR-GNNs, as well as the differences between AC-GNNs and ACR-GNNs, can be observed when we learn from examples. In particular, we show that on synthetic graph data conforming to $FOC_2$ formulas, AC-GNNs struggle to fit the training data while ACR-GNNs can generalize even to graphs of sizes not seen during training.

## 2  GRAPH NEURAL NETWORKS

In this section we describe the architecture of AC-GNNs and introduce other related notions. We concentrate on the problem of Boolean node classification: given a (simple, undirected) graph $G = (V, E)$ in which each vertex $v \in V$ has an associated feature vector $\boldsymbol{x}_v$, we wish to classify each graph node as true or false; in this paper, we assume that these feature vectors are one-hot

encodings of node colors in the graph, from a finite set of colors. The *neighborhood* $\mathcal{N}_G(v)$ of a node $v \in V$ is the set $\{u \mid \{v, u\} \in E\}$.

The basic architecture for GNNs, and the one studied in recent studies on GNN expressibility (Morris et al., 2019; Xu et al., 2019), consists of a sequence of *layers* that combine the feature vectors of every node with the multiset of feature vectors of its neighbors. Formally, let $\{\text{AGG}^{(i)}\}_{i=1}^{L}$ and $\{\text{COM}^{(i)}\}_{i=1}^{L}$ be two sets of *aggregation* and *combination* functions. An *aggregate-combine GNN* (AC-GNN) computes vectors $\boldsymbol{x}_v^{(i)}$ for every node $v$ of the graph $G$, via the recursive formula

$$\boldsymbol{x}_v^{(i)} = \text{COM}^{(i)}\left(\boldsymbol{x}_v^{(i-1)}, \text{AGG}^{(i)}\left(\{\!\!\{\boldsymbol{x}_u^{(i-1)} \mid u \in \mathcal{N}_G(v)\}\!\!\}\right)\right), \quad \text{for } i = 1, \ldots, L \qquad (1)$$

where each $\boldsymbol{x}_v^{(0)}$ is the initial feature vector $\boldsymbol{x}_v$ of $v$. Finally, each node $v$ of $G$ is classified according to a Boolean *classification* function CLS applied to $\boldsymbol{x}_v^{(L)}$. Thus, an AC-GNN with $L$ layers is defined as a tuple $\mathcal{A} = \left(\{\text{AGG}^{(i)}\}_{i=1}^{L}, \{\text{COM}^{(i)}\}_{i=1}^{L}, \text{CLS}\right)$, and we denote by $\mathcal{A}(G, v)$ the class (i.e., true or false) assigned by $\mathcal{A}$ to each node $v$ in $G$.[1]

There are many possible aggregation, combination, and classification functions, which produce different classes of GNNs (Hamilton et al., 2017; Kipf & Welling, 2017; Morris et al., 2019; Xu et al., 2019). A simple, yet common choice is to consider the sum of the feature vectors as the aggregation function, and a combination function as

$$\text{COM}^{(i)}(\boldsymbol{x}_1, \boldsymbol{x}_2) = f\left(\boldsymbol{x}_1 \boldsymbol{C}^{(i)} + \boldsymbol{x}_2 \boldsymbol{A}^{(i)} + \boldsymbol{b}^{(i)}\right), \qquad (2)$$

where $\boldsymbol{C}^{(i)}$ and $\boldsymbol{A}^{(i)}$ are matrices of parameters, $\boldsymbol{b}^{(i)}$ is a *bias* vector, and $f$ is a *non-linearity* function, such as relu or sigmoid. We call *simple* an AC-GNN using these functions. Furthermore, we say that an AC-GNN is *homogeneous* if all $\text{AGG}^{(i)}$ are the same and all $\text{COM}^{(i)}$ are the same (share the same parameters across layers). In most of our positive results we construct simple and homogeneous GNNs, while our negative results hold in general (i.e., for GNNs with arbitrary aggregation, combining, and classification functions).

The *Weisfeiler-Lehman* (*WL*) test is a powerful heuristic used to solve the graph isomorphism problem (Weisfeiler & Leman, 1968), or, for our purposes, to determine whether the neighborhoods of two nodes in a graph are structurally close or not. Due to space limitations, we refer to (Cai et al., 1992) for a formal definition of the underlying algorithm, giving only its informal description: starting from a colored graph, the algorithm iteratively assigns, for a certain number of *rounds*, a new color to every node in the graph; this is done in such a way that the color of a node in each round has a one to one correspondence with its own color and the multiset of colors of its neighbors in the previous round. An important observation is that the rounds of the WL algorithm can be seen as the layers of an AC-GNN whose aggregation and combination functions are all injective (Morris et al., 2019; Xu et al., 2019). Furthermore, as the following proposition states, an AC-GNN classification can never contradict the WL test.

**Proposition 2.1 (Morris et al., 2019; Xu et al., 2019).** *If the WL test assigns the same color to two nodes in a graph, then every AC-GNN classifies either both nodes as* true *or both nodes as* false.

## 3 CONNECTION BETWEEN GNNS AND LOGIC

### 3.1 LOGICAL NODE CLASSIFIERS

Our study relates the power of GNNs to that of classifiers expressed in first order (FO) predicate logic over (undirected) graphs where each vertex has a unique color (recall that we call these classifiers *logical classifiers*). To illustrate the idea of logical node classifiers, consider the formula

$$\alpha(x) := \text{Red}(x) \wedge \exists y \left(E(x, y) \wedge \text{Blue}(y)\right) \wedge \exists z \left(E(x, z) \wedge \text{Green}(z)\right). \qquad (3)$$

---

[1]For graph classification, which we do not consider in this paper, the classification function CLS inputs the multiset $\{\!\!\{\boldsymbol{x}_v^{(L)} \mid v \in V\}\!\!\}$ and outputs a class for the whole graph. Such a function is often called *readout* in previous work (Morris et al., 2019; Xu et al., 2019). In this paper, however, we use the term *readout* to refer to intermediate global operations performed while computing features for nodes (see Section 5).

This formula has one *free variable*, $x$, which is not bounded by any quantifier of the form $\exists$ or $\forall$, and two *quantified* variables $y$ and $z$. In general, formulas with one free variable are evaluated over nodes of a given graph. For example, the above formula evaluates to true exactly in those nodes $v$ whose color is Red and that have both a Blue and a Green neighbor. In this case, we say that node $v$ of $G$ satisfies $\alpha$, and denote this by $(G, v) \models \alpha$.

Formally, a logical (node) classifier is given by a formula $\varphi(x)$ in FO logic with exactly one free variable. This formula classifies as true those nodes $v$ in $G$ such that $(G, v) \models \varphi$, while all other nodes (i.e., those with $(G, v) \not\models \varphi$) are classified as false. We say that a GNN classifier captures a logical classifier when both classifiers coincide over every node in every possible input graph.

**Definition 3.1.** *A GNN classifier $\mathcal{A}$ captures a logical classifier $\varphi(x)$ if for every graph $G$ and node $v$ in $G$, it holds that $\mathcal{A}(G, v) = \mathsf{true}$ if and only if $(G, v) \models \varphi$.*

## 3.2 Logic FOC$_2$

Logical classifiers are useful as a declarative formalism, but as we will see, they are too powerful to compare them to AC-GNNs. Instead, for reasons we explain later we focus on classifiers given by formulas in FOC$_2$, the fragment of FO logic that only allows formulas with two variables, but in turn permits to use *counting quantifiers*.

Let us briefly introduce FOC$_2$ and explain why it is a restriction of FO logic. The first remark is that reducing the number of variables used in formulas drastically reduces their expressive power. Consider for example the following FO formula expressing that $x$ is a red node, and there is another node, $y$, that is not connected to $x$ and that has at least two blue neighbors, $z_1$ and $z_2$:

$$\beta(x) := \text{Red}(x) \land \exists y \big( \neg E(x, y) \land \exists z_1 \exists z_2 \big[ E(y, z_1) \land E(y, z_2) \land z_1 \neq z_2 \land \text{Blue}(z_1) \land \text{Blue}(z_2) \big] \big).$$

The formula $\beta(x)$ uses four variables, but it is possible to find an equivalent one with just three: the trick is to *reuse* variable $x$ and replace every occurrence of $z_2$ in $\beta(x)$ by $x$. However, this is as far as we can go with this trick: $\beta(x)$ does not have an equivalent formula with less than three variables. In the same way, the formula $\alpha(x)$ given in Equation (3) can be expressed using only two variables, $x$ and $y$, simply by reusing $y$ in place of $z$.

That being said, it is possible to extend the logic so that some node properties, such as the one defined by $\beta(x)$, can be expressed with even less variables. To this end, consider the counting quantifier $\exists^{\geq N}$ for every positive integer $N$. Analogously to how the quantifier $\exists$ expresses the existence of a node satisfying a property, the quantifier $\exists^{\geq N}$ expresses the existence of *at least $N$ different nodes* satisfying a property. For example, with $\exists^{\geq 2}$ we can express $\beta(x)$ by using only two variables by means of the classifier

$$\gamma(x) := \text{Red}(x) \land \exists y \big( \neg E(x, y) \land \exists^{\geq 2} x \big[ E(y, x) \land \text{Blue}(x) \big] \big). \tag{4}$$

Based on this idea, the logic FOC$_2$ allows for formulas using all FO constructs and counting quantifiers, but restricted to only two variables. Note that, in terms of their logical expressiveness, we have that FOC$_2$ is strictly less expressive than FO (as counting quantifiers can always be mimicked in FO by using more variables and disequalities), but is strictly more expressive than FO$_2$, the fragment of FO that allows formulas to use only two variables (as $\beta(x)$ belongs to FOC$_2$ but not to FO$_2$).

The following result establishes a classical connection between FOC$_2$ and the WL test. Together with Proposition 2.1, this provides a justification for our choice of logic FOC$_2$ for measuring the expressiveness of AC-GNNs.

**Proposition 3.2 (Cai et al., 1992).** *For any graph $G$ and nodes $u, v$ in $G$, the WL test colors $v$ and $u$ the same after any number of rounds iff $u$ and $v$ are classified the same by all FOC$_2$ classifiers.*

## 3.3 FOC$_2$ and AC-GNN classifiers

Having Propositions 2.1 and 3.2, one may be tempted to combine them and claim that every FOC$_2$ classifier can be captured by an AC-GNN. Yet, this is not the case as shown in Proposition 3.3 below. In fact, while it is true that two nodes are declared indistinguishable by the WL test if and only if they are indistinguishable by all FOC$_2$ classifiers (Proposition 3.2), and if the former holds then such nodes cannot be distinguished by AC-GNNs (Proposition 2.1), this by no means tells us that every FOC$_2$ classifier can be expressed as an AC-GNN.

**Proposition 3.3.** *There is an FOC$_2$ classifier that is not captured by any AC-GNN.*

One such FOC$_2$ classifier is $\gamma(x)$ in Equation (4), but there are infinitely many and even simpler FOC$_2$ formulas that cannot be captured by AC-GNNs. Intuitively, the main problem is that an AC-GNN has only a fixed number $L$ of layers and hence the information of local aggregations cannot travel further than at distance $L$ of every node along edges in the graph. For instance, the red node in $\gamma(x)$ may be farther away than the node with the blue neighbours, which means that AC-GNNs would never be able to connect this information. Actually, both nodes may even be in different connected components of a graph, in which case no number of layers would suffice.

The negative result of Proposition 3.3 opens up the following important questions.

1. What kind of FOC$_2$ classifiers can be captured by AC-GNNs?
2. Can we capture FOC$_2$ classifiers with GNNs using a simple extension of AC-GNNs?

We provide answers to these questions in the next two sections.

## 4  THE EXPRESSIVE POWER OF AC-GNNS

Towards answering our first question, we recall that the problem with AC-GNN classifiers is that they are local, in the sense that they cannot see across a distance greater than their number of layers. Thus, if we want to understand which logical classifiers this architecture is capable of expressing, we must consider logics built with similar limitations in mind. And indeed, in this section we show that AC-GNNs capture any FOC$_2$ classifier as long as we further restrict the formulas so that they satisfy such a locality property. This happens to be a well-known restriction of FOC$_2$, and corresponds to graded modal logic (de Rijke, 2000) or, equivalently, to description logic $\mathcal{ALCQ}$ (Baader et al., 2003), which is fundamental for knowledge representation: for instance, the OWL 2 Web Ontology Language (Motik et al., 2012; W3C OWL Working Group, 2012) relies on $\mathcal{ALCQ}$.

The idea of graded modal logic is to force all subformulas to be *guarded* by the edge predicate $E$. This means that one cannot express in graded modal logic arbitrary formulas of the form $\exists y \varphi(y)$, i.e., whether there is some node that satisfies property $\varphi$. Instead, one is allowed to check whether some neighbor $y$ of the node $x$ where the formula is being evaluated satisfies $\varphi$. That is, we are allowed to express the formula $\exists y \left( E(x,y) \wedge \varphi(y) \right)$ in the logic as in this case $\varphi(y)$ is guarded by $E(x,y)$. We can define this fragment of FO logic using FO syntax as follows. A graded modal logic formula is either $\mathrm{Col}(x)$, for Col a node color, or one of the following, where $\varphi$ and $\psi$ are graded modal logic formulas and $N$ is a positive integer:

$$\neg \varphi(x), \quad \varphi(x) \wedge \psi(x), \quad \exists^{\geq N} y \left( E(x,y) \wedge \varphi(y) \right).$$

Notice then that the formula $\delta(x) := \mathrm{Red}(x) \wedge \exists y \left( E(x,y) \wedge \mathrm{Blue}(y) \right)$ is in graded modal logic, but the logical classifier $\gamma(x)$ in Equation (4) is not, because the use of $\neg E(x,y)$ as a guard is disallowed. As required, we can now show that AC-GNNs can indeed capture all graded modal logic classifiers.

**Proposition 4.1.** *Each graded modal logic classifier is captured by a simple homogeneous AC-GNN.*

The key idea of the construction is that the vectors' dimensions used by the AC-GNN to label nodes, represent the sub-formulas of the captured classifier. Thus, if a feature in a node is 1 then the node satisfies the corresponding sub-formula, and the opposite holds after evaluating $L$ layers, where $L$ is the "quantifier depth" of the classifier (which does not depend on the graph). The construction uses simple, homogeneous AC-GNNs with the truncated relu non-linearity $\max(0, \min(x, 1))$. The formal proof of Proposition 4.1, as well as other formal statements, can be found in the Appendix. An interesting question that we leave as future work is to investigate whether the same kind of construction can be done with AC-GNNs using different aggregate and combine operators than the ones we consider here; for instance, using max instead of sum to aggregate the feature vectors of the neighbors, or using other non-linearity such as sigmoid, etc.

The relationship between AC-GNNs and graded modal logic goes further: we can show that graded modal logic is the "largest" class of logical classifiers captured by AC-GNNs. This means that the only FO formulas that AC-GNNs are able to learn accurately are those in graded modal logic.

**Theorem 4.2.** *A logical classifier is captured by AC-GNNs if and only if it can be expressed in graded modal logic.*

The backward direction of this theorem is Proposition 4.1, while the proof of the forward direction is based on a recently communicated extension of deep results in finite model theory (Otto, 2019). We point out that the forward direction holds no matter which aggregate and combine operators are considered, i.e., this is a limitation of the architecture for AC-GNNs, not of the specific functions that one chooses to update the features.

## 5 GNNs FOR CAPTURING FOC$_2$

### 5.1 GNNs WITH GLOBAL READOUTS

In this section we tackle our second question: which kind of GNN architecture we need to capture all FOC$_2$ classifiers? Recall that the main shortcoming of AC-GNNs for expressing such classifiers is their local behavior. A natural way to break such a behavior is to allow for a global feature computation on each layer of the GNN. This is called a *global attribute* computation in the framework of Battaglia et al. (2018). Following the recent GNN literature (Gilmer et al., 2017; Morris et al., 2019; Xu et al., 2019), we refer to this global operation as a *readout*.

Formally, an *aggregate-combine-readout GNN* (*ACR-GNN*) extends AC-GNNs by specifying *readout* functions $\{\text{READ}^{(i)}\}_{i=1}^{L}$, which aggregate the current feature vectors of all the nodes in a graph. Then, the vector $\boldsymbol{x}_v^{(i)}$ of each node $v$ in $G$ on each layer $i$, is computed by the following formula, generalizing Equation (1):

$$\boldsymbol{x}_v^{(i)} = \text{COM}^{(i)}\Big(\boldsymbol{x}_v^{(i-1)}, \text{AGG}^{(i)}\big(\{\!\{\boldsymbol{x}_u^{(i-1)} \mid u \in \mathcal{N}_G(v)\}\!\}\big), \text{READ}^{(i)}\big(\{\!\{\boldsymbol{x}_u^{(i-1)} \mid u \in G\}\!\}\big)\Big). \quad (5)$$

Intuitively, every layer in an ACR-GNN first computes (i.e., "reads out") the aggregation over all the nodes in $G$; then, for every node $v$, it computes the aggregation over the neighbors of $v$; and finally it combines the features of $v$ with the two aggregation vectors. All the notions about AC-GNNs extend to ACR-GNNs in a straightforward way; for example, a *simple* ACR-GNN uses the sum as the function $\text{READ}^{(i)}$ in each layer, and the combination function $\text{COM}^{(i)}(\boldsymbol{x}_1, \boldsymbol{x}_2, \boldsymbol{x}_3) = f\big(\boldsymbol{x}_1 \boldsymbol{C}^{(i)} + \boldsymbol{x}_2 \boldsymbol{A}^{(i)} + \boldsymbol{x}_3 \boldsymbol{R}^{(i)} + \boldsymbol{b}^{(i)}\big)$ with a matrix $\boldsymbol{R}^{(i)}$, generalizing Equation (2).

### 5.2 ACR-GNNs AND FOC$_2$

To see how a readout function could help in capturing non-local properties, consider again the logical classifier $\gamma(x)$ in Equation (4), that assigns true to every red node $v$ as long as there is another node not connected with $v$ having two blue neighbors. We have seen that AC-GNNs cannot capture this classifier. However, using a single readout plus local aggregations one can implement this classifier as follows. First, define by $B$ the property "having at least 2 blue neighbors". Then an ACR-GNN that implements $\gamma(x)$ can (1) use one aggregation to store in the local feature of every node if the node satisfies $B$, then (2) use a readout function to count how many nodes satisfying $B$ exist in the whole graph, and (3) use another local aggregation to count how many neighbors of every node satisfiy $B$. Then $\gamma$ is obtained by classifying as true every red node having less neighbors satisfying $B$ than the total number of nodes satisfying $B$ in the whole graph. It turns out that the usage of readout functions is enough to capture all non-local properties of FOC$_2$ classifiers.

**Theorem 5.1.** *Each FOC$_2$ classifier can be captured by a simple homogeneous ACR-GNN.*

The construction is similar to that of Proposition 4.1 and uses simple, homogeneous ACR-GNNs—that is, the readout function is just the sum of all the local node feature vectors. Moreover, the readout functions are only used to deal with subformulas asserting the existence of a node that is not connected to the current node in the graph, just as we have done for classifier $\gamma(x)$. As an intermediate step in the proof, we use a characterization of FOC$_2$ using an extended version of graded modal logic, which was obtained by Lutz et al. (2001). We leave as a challenging open problem whether FOC$_2$ classifiers are exactly the logical classifiers captured by ACR-GNNs.

## 5.3 COMPARING THE NUMBER OF READOUT LAYERS

The proof of Theorem 5.1 constructs GNNs whose number of layers depends on the formula being captured—that is, readout functions are used unboundedly many times in ACR-GNNs for capturing different $FOC_2$ classifiers. Given that a global computation can be costly, one might wonder whether this is really needed, or if it is possible to cope with all the complexity of such classifiers by performing only few readouts. We next show that actually just one readout is enough. However, this reduction in the number of readouts comes at the cost of severely complicating the resulting GNN.

Formally, an *aggregate-combine GNN with final readout* (AC-FR-GNN) results out of using any number of layers as in the AC-GNN definition, together with a final layer that uses a readout function, according to Equation (5).

**Theorem 5.2.** *Each $FOC_2$ classifier is captured by an AC-FR-GNN.*

The AC-FR-GNN in the proof of this theorem is not based on the idea of evaluating the formula incrementally along layers, as in the proofs of Proposition 4.1 and Theorem 5.1, and it is not simple (note that AC-FR-GNNs are never homogeneous). Instead, it is based on a refinement of the GIN architecture proposed by Xu et al. (2019) to obtain as much information as possible about the local neighborhood in graphs, followed by a readout and combine functions that use this information to deal with non-local constructs in formulas. The first component we build is an AC-GNN that computes an invertible function mapping each node to a number representing its neighborhood (how big is this neighborhood depends on the classifier to be captured). This information is aggregated so that we know for each different type of a neighborhood how many times it appears in the graph. We then use the combine function to evaluate $FOC_2$ formulas by decoding back the neighborhoods.

## 6 EXPERIMENTAL RESULTS

We perform experiments with synthetic data to empirically validate our results. The motivation of this section is to show that the theoretical expressiveness of ACR-GNNs, as well as the differences between AC- and ACR-GNNs, can actually be observed when we learn from examples. We perform two sets of experiments: experiments to show that ACR-GNNs can learn a very simple $FOC_2$ node classifier that AC-GNNs cannot learn, and experiments involving complex $FOC_2$ classifiers that need more intermediate readouts to be learned. We implemented our experiments in the PyTorch Geometric library (Fey & Lenssen, 2019). Besides testing simple AC-GNNs, we also tested the GIN network proposed by Xu et al. (2019) (we consider the implementation by Fey & Lenssen (2019) and adapted it to classify nodes). Our experiments use synthetic graphs, with five initial colors encoded as one-hot features, divided in three sets: train set with 5k graphs of size up to 50-100 nodes, test set with 500 graphs of size similar to the train set, and another test set with 500 graphs of size bigger than the train set. We tried several configurations for the aggregation, combination and readout functions, and report the accuracy on the best configuration. Accuracy in our experiments is computed as the total number of nodes correctly classified among all nodes in all the graphs in the dataset. In every case we run up to 20 epochs with the Adam optimizer. More details on the experimental setting, data, and code can be found in the Appendix. We finally report results on a real benchmark (PPI) where we did not observe an improvement of ACR-GNNs over AC-GNNs.

**Separating AC-GNNs and ACR-GNNs**   We consider a very simple $FOC_2$ formula defined by $\alpha(x) := \text{Red}(x) \wedge \exists y \, \text{Blue}(y)$, which is satisfied by every red node in a graph provided that the graph contains at least one blue node. We tested with line-shaped graphs and Erdös-Renyi (E-R) random graphs with different connectivities. In every set (train and test) we consider 50% of graphs not containing any *blue* node, and 50% containing at least one *blue* node (around 20% of nodes are in the true class in every set). For both types of graphs, already single-layer ACR-GNNs showed perfect performance (ACR-1 in Table 1). This was what we expected given the simplicity of the property being checked. In contrast, AC-GNNs and GINs (shown in Table 1 as AC-$L$ and GIN-$L$, representing AC-GNNs and GINs with $L$ layers) struggle to fit the data. For the case of the line-shaped graph, they were not able to fit the train data even by allowing 7 layers. For the case of random graphs, the performance with 7 layers was considerably better. In a closer look at the performance for different connectivities of E-R graphs, we found an improvement for AC-GNNs when we train them with more dense graphs (details in the Appendix). This is consistent with the fact that AC-GNNs are able to move information of local aggregations to distances up to their

|  | Line Train | Line Test | | E-R Train | E-R Test | |
|---|---|---|---|---|---|---|
|  |  | same-size | bigger |  | same-size | bigger |
| AC-5 | 0.887 | 0.886 | 0.892 | 0.951 | 0.949 | 0.929 |
| AC-7 | 0.892 | 0.892 | 0.897 | 0.967 | 0.965 | 0.958 |
| GIN-5 | 0.861 | 0.861 | 0.867 | 0.830 | 0.831 | 0.817 |
| GIN-7 | 0.863 | 0.864 | 0.870 | 0.818 | 0.819 | 0.813 |
| ACR-1 | 1.000 | 1.000 | 1.000 | 1.000 | 1.000 | 1.000 |

Table 1: Results on synthetic data for nodes labeled by classifier $\alpha(x) := \text{Red}(x) \wedge \exists y \, \text{Blue}(y)$

|  | $\alpha_1$ Train | $\alpha_1$ Test | | $\alpha_2$ Train | $\alpha_2$ Test | | $\alpha_3$ Train | $\alpha_3$ Test | |
|---|---|---|---|---|---|---|---|---|---|
|  |  | same-size | bigger |  | same-size | bigger |  | same-size | bigger |
| AC | 0.839 | 0.826 | 0.671 | 0.694 | 0.695 | 0.667 | 0.657 | 0.636 | 0.632 |
| GIN | 0.567 | 0.566 | 0.536 | 0.689 | 0.693 | 0.672 | 0.656 | 0.643 | 0.580 |
| AC-FR-2 | 1.000 | 1.000 | 1.000 | 0.863 | 0.860 | 0.694 | 0.788 | 0.775 | 0.770 |
| AC-FR-3 | 1.000 | 1.000 | 0.825 | 0.840 | 0.823 | 0.604 | 0.787 | 0.767 | 0.771 |
| ACR-1 | 1.000 | 1.000 | 1.000 | 0.827 | 0.834 | 0.726 | 0.760 | 0.762 | 0.773 |
| ACR-2 | 1.000 | 1.000 | 1.000 | 0.895 | 0.897 | 0.770 | 0.800 | 0.799 | 0.771 |
| ACR-3 | 1.000 | 1.000 | 1.000 | 0.903 | 0.902 | 0.836 | 0.817 | 0.802 | 0.748 |

Table 2: Results on E-R synthetic data for nodes labeled by classifiers $\alpha_i(x)$ in Equation (6)

number of layers. This combined with the fact that random graphs that are more dense make the maximum distances between nodes shorter, may explain the boost in performance for AC-GNNs.

**Complex FOC$_2$ properties** In the second experiment we consider classifiers $\alpha_i(x)$ constructed as

$$\alpha_0(x) := \text{Blue}(x), \qquad \alpha_{i+1}(x) := \exists^{[N,M]} y\big(\alpha_i(y) \wedge \neg E(x,y)\big), \qquad (6)$$

where $\exists^{[N,M]}$ stands for "there exist between $N$ and $M$ nodes" satisfying a given property. Observe that each $\alpha_i(x)$ is in FOC$_2$, as $\exists^{[N,M]}$ can be expressed by combining $\exists^{\geq N}$ and $\neg\exists^{\geq M+1}$. We created datasets with E-R dense graphs and labeled them according to $\alpha_1(x)$, $\alpha_2(x)$, and $\alpha_3(x)$, ensuring in each case that approximately half of all nodes in our dataset satisfy every property. Our experiments show that when increasing the depth of the formula (existential quantifiers with negations inside other existential quantifiers) more layers are needed to increase train and test accuracy (see Table 2). We report ACR-GNNs performance up to 3 layers (ACR-$L$ in Table 2) as beyond that we did not see any significant improvement. We also note that for the bigger test set, AC-GNNs and GINs are unable to substantially depart from a trivial baseline of 50%. We tested these networks with up to 10 layers but only report the best results on the bigger test set. We also test AC-FR-GNNs with two and three layers (AC-FR-$L$ in Table 2). As we expected, although theoretically using a single readout gives the same expressive power as using several of them (Theorem 5.2), in practice more than a single readout can actually help the learning process of complex properties.

**PPI** We also tested AC- and ACR-GNNs on the Protein-Protein Interaction (PPI) benchmark (Zitnik & Leskovec, 2017). We chose PPI since it is a node classification benchmark with different graphs in the train set (as opposed to other popular benchmarks for node classification such as Core or Citeseer that have a single graph). Although the best results for both classes of GNNs on PPI were quite high (AC: 97.5 F1, ACR: 95.4 F1 in the test set), we did not observe an improvement when using ACR-GNNs. Chen et al. (2019) recently observed that commonly used benchmarks are inadequate for testing advanced GNN variants, and ACR-GNNs might be suffering from this fact.

## 7 FINAL REMARKS

Our results show the theoretical advantages of mixing local and global information when classifying nodes in a graph. Recent works have also observed these advantages in practice, e.g., Deng et al.

(2018) use global-context aware local descriptors to classify objects in 3D point clouds, You et al. (2019) construct node features by computing shortest-path distances to a set of distant anchor nodes, and Haonan et al. (2019) introduced the idea of a "star node" that stores global information of the graph. As mentioned before, our work is close in spirit to that of Xu et al. (2019) and Morris et al. (2019) establishing the correspondence between the WL test and GNNs. In contrast to our work, they focus on graph classification and do not consider the relationship with logical classifiers.

Regarding our results on the links between AC-GNNs and graded modal logic (Theorem 4.2), we point out that very recent work of Sato et al. (2019) establishes close relationships between GNNs and certain classes of *distributed local algorithms*. These in turn have been shown to have strong correspondences with modal logics (Hella et al., 2015). Hence, variants of our Proposition 4.1 could be obtained by combining these two lines of work (but it is not clear if this combination would yield AC-GNNs that are *simple*). However, these works do not investigate the impact of having *non-local computations* (such as the readouts that we consider), hence our results on the relationships between FO an ACR-GNNs (Theorem 5.1 and 5.2) do not follow from these.

Morris et al. (2019) also studied $k$-GNNs, which are inspired by the $k$-dimensional WL test. In $k$-GNNs, graphs are considered as structures connecting $k$-tuples of nodes instead of just pairs of them. We plan to study how our results on logical classifiers relate to $k$-GNNs, in particular, with respect to the logic $FOC_k$ that extends $FOC_2$ by allowing formulas with $k$ variables, for each fixed $k > 1$. Recent work has also explored the extraction of finite state representations from recurrent neural networks as a way of explaining them (Weiss et al., 2018; Koul et al., 2019; Oliva & Lago-Fernández, 2019). We would like to study how our results can be applied for extracting logical formulas from GNNs as possible explanations for their computations.

### ACKNOWLEDGMENTS

This work was partly funded by the Millennium Institute for Foundational Research on Data[2].

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

APPENDIX

## A    PROOF OF PROPOSITION 3.3

We first recall the proposition.

**Proposition 3.3.** *There is an $FOC_2$ classifier that is not captured by any AC-GNN.*

*Proof.* Consider the following $FOC_2$ node property $\alpha(v) := \text{Red}(v) \wedge \exists x \, \text{Green}(x)$. We will show by contradiction that there is no AC-GNN that captures $\alpha$, no matter which aggregation, combining, and final classification functions are allowed. Indeed, assume that $\mathcal{A}$ is an AC-GNN capturing $\alpha$, and let $L$ be its number of layers. Consider the graph $G$ that is a chain of $L + 2$ nodes colored Red, and consider the first node $v_0$ in that chain. Since $\mathcal{A}$ captures $\alpha$, and since $(G, v_0) \not\models \alpha$, we have that $\mathcal{A}$ labels $v_0$ with false, i.e., $\mathcal{A}(G, v_0) = \text{false}$. Now, consider the graph $G'$ obtained from $G$ by coloring the last node in the chain with Green (instead of Red). Then one can easily show that $\mathcal{A}$ again labels $v_0$ by false in $G'$. But we have $(G', v_0) \models \alpha$, a contradiction.

The above proof relies on the following weakness of AC-GNNs: if the number of layers is fixed (i.e., does not depend on the input graph), then the information of the color of a node $v$ cannot travel further than at distance $L$ from $v$. Nevertheless, we can show that the same holds even when we consider AC-GNNs that dispose of an arbitrary number of layers (for instance, one may want to run a homogeneous AC-GNN for $f(|E|)$ layers for each graph $G = (V, E)$, for a fixed function $f$). Assume again by way of contradiction that $\mathcal{A}$ is such an extended AC-GNN capturing $\alpha$. Consider the graph $G$ consisting of two disconnected nodes $v, u$, with $v$ colored Red and $y$ colored Green. Then, since $(G, v) \models \alpha$, we have $\mathcal{A}(G, v) = \text{true}$. Now consider the graph $G'$ obtained from $G$ by changing the color of $u$ from Green to Red. Observe that, since the two nodes are not connected, we will again have $\mathcal{A}(G', v) = \text{true}$, contradicting the fact that $(G', v) \not\models \alpha$ and that $\mathcal{A}$ is supposed to capture $\alpha$.

By contrast, it is easy to see that this formula can be done with only one intermediate readout, using the technique in the proof of Theorem 5.1. $\qquad\square$

## B    PROOF OF PROPOSITION 4.1

We first recall the proposition.

**Proposition 4.1.** *Each graded modal logic classifier is captured by a simple homogeneous AC-GNN.*

We first define formally the semantics of the graded modal logic (de Rijke, 2000) over simple undirected node-colored graphs (de Rijke, 2000), assuming the FO syntax introduced in the paper.

**Definition B.1.** *We define when a node $v$ in a graph $G$ satisfies a graded modal logic formula $\varphi(x)$, written as $v \models \varphi$ in $G$ (where "in $G$" may be omitted when clear), recursively as follows:*

- *if $\varphi(x) = \text{Col}(x)$, then $v \models \varphi$ if and only if Col is the color of $v$ in $G$,*

- *if $\varphi(x) = \varphi'(x) \wedge \varphi''(x)$, then $v \models \varphi$ if and only if $v \models \varphi'$ and $v \models \varphi''$, and similarly with $\neg\varphi'(x)$, and*

- *if $\varphi(x) = \exists^{\geq N}(E(x, y) \wedge \varphi'(y))$, then $v \models \varphi$ if and only if the set of nodes $\{u \mid u \in \mathcal{N}_G(v)$ and $v \models \varphi'\}$ has cardinality at least $N$.*

We can now proceed to the proof of the proposition.

*Proof of Proposition 4.1.* Let $\varphi(x)$ be a graded modal logic formula. We will construct an AC-GNN $\mathcal{A}_\varphi$ that is further simple and homogeneous. Let $\text{sub}(\varphi) = (\varphi_1, \varphi_2, \ldots, \varphi_L)$ be an enumeration of the sub-formulas of $\varphi$ such that if $\varphi_k$ is a subformula of $\varphi_\ell$ then $k \leq \ell$. The idea of the construction of $\mathcal{A}_\varphi$ is to have feature vectors in $\mathbb{R}^L$ such that every component of those vectors represents a different formula in $\text{sub}(\varphi)$. Then $\mathcal{A}_\varphi$ will update the feature vector $\boldsymbol{x}_v^{(i)}$ of node $v$ ensuring that component $\ell$ of $\boldsymbol{x}_v^{(\ell)}$ gets a value 1 if and only if the formula $\varphi_\ell$ is satisfied in node $v$.

We note that $\varphi = \varphi_L$ and thus, the last component of each feature vector after evaluating $L$ layers in every node gets a value $1$ if and only if the node satisfies $\varphi$. We will then be able to use a final classification function CLS that simply extracts that particular component.

Formally, the simple homogeneous AC-GNN $\mathcal{A}_\varphi$ has $L$ layers and uses the aggregation and combine functions

$$
\begin{aligned}
\text{AGG}(X) &= \sum_{\boldsymbol{x} \in X} \boldsymbol{x}, \\
\text{COM}(\boldsymbol{x}, \boldsymbol{y}) &= \sigma(\boldsymbol{x}\boldsymbol{C} + \boldsymbol{y}\boldsymbol{A} + \boldsymbol{b}),
\end{aligned}
$$

where $\boldsymbol{A}, \boldsymbol{C} \in \mathbb{R}^{L \times L}$, and $\boldsymbol{b} \in \mathbb{R}^L$ are defined next, and $\sigma$ is the truncated ReLU activation defined by $\sigma(x) = \min(\max(0, x), 1)$. The entries of the $\ell$-th columns of $\boldsymbol{A}, \boldsymbol{C}$, and $\boldsymbol{b}$ depend on the sub-formulas of $\varphi$ as follows:

*Case 0.* if $\varphi_\ell(x) = \text{Col}(x)$ with Col one of the (base) colors, then $C_{\ell\ell} = 1$,

*Case 1.* if $\varphi_\ell(x) = \varphi_j(x) \wedge \varphi_k(x)$ then $C_{j\ell} = C_{k\ell} = 1$ and $b_\ell = -1$,

*Case 2.* if $\varphi_\ell(x) = \neg\varphi_k(x)$ then $C_{k\ell} = -1$ and $b_\ell = 1$,

*Case 3.* if $\varphi_\ell(x) = \exists^{\geq N}(E(x, y) \wedge \varphi_k(y))$ then $A_{k\ell} = 1$ and $b_\ell = -N + 1$,

and all other values in the $\ell$-th columns of $\boldsymbol{A}, \boldsymbol{C}$, and $\boldsymbol{b}$ are $0$.

We now prove that $\mathcal{A}_\varphi$ indeed captures $\varphi$. Let $G = (V, E)$ be a colored graph. For every node $v$ in $G$ we consider the initial feature vector $\boldsymbol{x}_v^{(0)} = (x_1, \ldots, x_L)$ such that $x_\ell = 1$ if sub-formula $\varphi_\ell$ is the initial color assigned to $v$, and $x_\ell = 0$ otherwise. By definition, AC-GNN $\mathcal{A}_\varphi$ will iterate the aggregation and combine functions defined above for $L$ rounds ($L$ layers) to produce feature vectors $\boldsymbol{x}_v^{(i)}$ for every node $v \in G$ and $\ell = 1, \ldots, L$ as follows:

$$
\begin{aligned}
\boldsymbol{x}_v^{(i)} &= \text{COM}(\boldsymbol{x}_v^{(i-1)}, \text{AGG}(\{\!\!\{\boldsymbol{x}_u^{(i-1)} \mid u \in \mathcal{N}(v)\}\!\!\})) \\
&= \sigma\left(\boldsymbol{x}_v^{(i-1)}\boldsymbol{C} + \sum_{u \in \mathcal{N}(v)} \boldsymbol{x}_u^{(i-1)}\boldsymbol{A} + \boldsymbol{b}\right).
\end{aligned}
\tag{7}
$$

We next prove that for every $\varphi_\ell \in \text{sub}(\varphi)$, every $i \in \{\ell, \ldots, L\}$, and every node $v$ in $G$ it holds that

$$
(\boldsymbol{x}_v^{(i)})_\ell = 1 \text{ if } v \models \varphi_\ell, \text{ and } (\boldsymbol{x}_v^{(i)})_\ell = 0 \text{ otherwise},
\tag{8}
$$

where $(\boldsymbol{x}_v^{(i)})_\ell$ is the $\ell$-th component of $\boldsymbol{x}_v^{(i)}$—that is, the $\ell$-th component of $\boldsymbol{x}_v^{(i)}$ has a $1$ if and only if $v$ satisfies $\varphi_\ell$ in $G$. In the rest of the proof we will be continuously using the value of $(\boldsymbol{x}_v^{(i)})_\ell$ whose general expression is

$$
(\boldsymbol{x}_v^{(i)})_\ell = \sigma\left(\sum_{k=1}^{L} (\boldsymbol{x}_v^{(i-1)})_k C_{k\ell} + \sum_{u \in \mathcal{N}(v)} \sum_{k=1}^{L} (\boldsymbol{x}_u^{(i-1)})_k A_{k\ell} + b_\ell\right).
\tag{9}
$$

We proceed to prove (8) by induction on the number of sub-formulas of every $\varphi_\ell$. If $\varphi_\ell$ has one sub-formula, then $\varphi_\ell(x) = \text{Col}(x)$ with Col a base color. We next prove that $(\boldsymbol{x}_v^{(1)})_\ell = 1$ if and only if $v$ has Col as its initial color. Since $\varphi_\ell(x) = \text{Col}(x)$ we know that $C_{\ell\ell} = 1$ and $C_{k\ell} = 0$ for every $k \neq \ell$ (see Case 0 above). Moreover, we know that $b_\ell = 0$ and $A_{k\ell} = 0$ for every $k$. Then, from Equation (9) we obtain that

$$
(\boldsymbol{x}_v^{(1)})_\ell = \sigma\left(\sum_{k=1}^{L} (\boldsymbol{x}_v^{(0)})_k C_{k\ell} + \sum_{\{v,u\} \in E} \sum_{k=1}^{L} (\boldsymbol{x}_u^{(0)})_k A_{k\ell} + b_\ell\right) = \sigma((\boldsymbol{x}_v^{(0)})_\ell).
$$

Then, given that $(\boldsymbol{x}_v^{(0)})_\ell = 1$ if the initial color of $v$ is Col and $(\boldsymbol{x}_v^{(0)})_\ell = 0$ otherwise, we have that $(\boldsymbol{x}_v^{(1)})_\ell = 1$ if $(G, v) \models \varphi_\ell$ and $(\boldsymbol{x}_v^{(1)})_\ell = 0$ otherwise. From this it is easy to prove that for every $i \geq 1$ the vector $(\boldsymbol{x}_v^{(i)})_\ell$ satisfies the same property. Now assume that $\varphi_\ell$ has more than one

sub-formula, and assume that for every $\varphi_k$ with $k < \ell$ the property (8) holds. Let $i \geq \ell$. We are left to consider the following cases, corresponding to the cases for the shape of the formula above.

*Case 1.* Assume that $\varphi_\ell(x) = \varphi_j(x) \wedge \varphi_k(x)$. Then $C_{j\ell} = C_{k\ell} = 1$ and $b_\ell = -1$. Moreover, we have $C_{m\ell} = 0$ for every $m \neq j, k$ and $A_{n\ell} = 0$ for every $n$ (see Case 2 above). Then, from Equation (9) we obtain that

$$(\boldsymbol{x}_v^{(i)})_\ell \;=\; \sigma\bigg((\boldsymbol{x}_v^{(i-1)})_j + (\boldsymbol{x}_v^{(i-1)})_k - 1\bigg).$$

Since the number of each proper sub-formula of $\varphi_\ell$ is strictly less than both $\ell$ and $i$, by induction hypothesis we know that $(\boldsymbol{x}_v^{(i-1)})_j = 1$ if and only if $v \models \varphi_j$ and $(\boldsymbol{x}_v^{(i-1)})_j = 0$ otherwise. Similarly, $(\boldsymbol{x}_v^{(i-1)})_k = 1$ if and only if $v \models \varphi_k$ and $(\boldsymbol{x}_v^{(i-1)})_k = 0$ otherwise. Now, since $(\boldsymbol{x}_v^{(i)})_\ell = \sigma((\boldsymbol{x}_v^{(i-1)})_j + (\boldsymbol{x}_v^{(i-1)})_k - 1)$ we have that $(\boldsymbol{x}_v^{(i)})_\ell = 1$ if and only if $(\boldsymbol{x}_v^{(i-1)})_j + (\boldsymbol{x}_v^{(i-1)})_k - 1 \geq 1$ that can only happen if $(\boldsymbol{x}_v^{(i-1)})_j = (\boldsymbol{x}_v^{(i-1)})_k = 1$. Then $(\boldsymbol{x}_v^{(i)})_\ell = 1$ if and only if $v \models \varphi_j$ and $v \models \varphi_k$—that is, if and only if $v \models \varphi_\ell$ (since $\varphi_\ell(x) = \varphi_j(x) \wedge \varphi_k(x)$), and $(\boldsymbol{x}_v^{(i)})_\ell = 0$ otherwise. This is exactly what we wanted to prove.

*Case 2.* Assume that $\varphi_\ell(x) = \neg\varphi_k(x)$. Then $C_{k\ell} = -1$ and $b_\ell = 1$. Moreover, we have $C_{m\ell} = 0$ for every $m \neq k$ and $A_{n\ell} = 0$ for every $n$ (see Case 2 above). Then, from Equation (9) we obtain that

$$(\boldsymbol{x}_v^{(i)})_\ell \;=\; \sigma\bigg(-(\boldsymbol{x}_v^{(i-1)})_k + 1\bigg).$$

By induction hypothesis we know that $(\boldsymbol{x}_v^{(i-1)})_k = 1$ if and only if $v \models \varphi_k$ and $(\boldsymbol{x}_v^{(i-1)})_k = 0$ otherwise. Since $(\boldsymbol{x}_v^{(i)})_\ell = \sigma(-(\boldsymbol{x}_v^{(i-1)})_k + 1)$ we have that $(\boldsymbol{x}_v^{(i)})_\ell = 1$ if and only if $1 - (\boldsymbol{x}_v^{(i-1)})_k \geq 1$ that can only happen if $(\boldsymbol{x}_v^{(i-1)})_k = 0$. Then $(\boldsymbol{x}_v^{(i)})_\ell = 1$ if and only if $v \not\models \varphi_k$—that is, if and only if $v \models \neg\varphi_k$, which holds if and only if $v \models \varphi_\ell$, and $(\boldsymbol{x}_v^{(i)})_\ell = 0$ otherwise. This is exactly what we wanted to prove.

*Case 3.* Assume that $\varphi_\ell(x) = \exists^{\geq N}(E(x, y) \wedge \varphi_k(y))$. Then $A_{k\ell} = 1$ and $b_\ell = -N + 1$. Moreover for every $m$ we have that $C_{m\ell} = 0$ (see Case 3 above). Then, from Equation (9) we obtain that

$$(\boldsymbol{x}_v^{(i)})_\ell \;=\; \sigma\bigg(-N + 1 + \sum_{\{u,v\} \in E} (\boldsymbol{x}_u^{(i-1)})_k\bigg).$$

By induction hypothesis we know that $(\boldsymbol{x}_u^{(i-1)})_k = 1$ if and only if $v \models \varphi_k$ and $(\boldsymbol{x}_u^{(i-1)})_k = 0$ otherwise. Then we can write $(\boldsymbol{x}_v^{(i)})_\ell = \sigma(-N + 1 + m)$ where

$$m = |\{u \mid u \in \mathcal{N}(v) \text{ and } u \models \varphi_k\}|.$$

Thus, we have that $(\boldsymbol{x}_v^{(i)})_\ell = 1$ if and only if $m \geq N$, that is if and only if there exists at least $N$ nodes connected with $v$ that satisfy $\varphi_k$, and $(\boldsymbol{x}_v^{(i)})_\ell = 0$ otherwise. From that we obtain that $(\boldsymbol{x}_v^{(i)})_\ell = 1$ if and only if $v \models \varphi_\ell$ since $\varphi_\ell(x) = \exists^{\geq N}(E(x, y) \wedge \varphi_k(y))$, which is what we wanted to prove.

To complete the proof we only need to add a final classification after the $L$ iterations of the aggregate and combine layers that simply classifies a node $v$ as true if the component of $\boldsymbol{x}_v^{(L)}$ corresponding to $\varphi$ holds 1. □

## C    PROOF OF THEOREM 4.2

We first recall the theorem.

**Theorem 4.2.** *A logical classifier is captured by AC-GNNs if and only if it can be expressed in graded modal logic.*

Note that one direction follows immediately from Proposition 4.1, so we only need to show the following proposition.

**Proposition C.1.** *If a logical classifier $\alpha$ is not equivalent to any graded modal logic formula, then there is no AC-GNN that captures $\alpha$.*

To prove this proposition, we will need the following definition, which is standard in modal logics theory.

**Definition C.2.** *Let $G$ be a graph (simple, undirected and node-colored), $v$ be a node in $G$, and $L \in \mathbb{N}$. The* unravelling *of $v$ in $G$ at depth $L$, denoted by $\mathrm{Unr}_G^L(v)$, is the (simple undirected node-colored) graph that is the tree having*

- *a node $(v, u_1, \ldots, u_i)$ for each path $(v, u_1, \ldots, u_i)$ in $G$ with $i \leq L$,*

- *an edge between $(v, u_1, \ldots, u_{i-1})$ and $(v, u_1, \ldots, u_i)$ when $\{u_{i-1}, u_i\}$ is an edge in $G$ (assuming that $u_0$ is $v$), and*

- *each node $(v, u_1, \ldots, u_i)$ colored the same as $u_i$ in $G$.*

We then observe the following.

**Observation C.3.** *Let $G$ and $G'$ be two graphs, and $v$ and $v'$ be two nodes in $G$ and $G'$, respectively. Then for every $L \in \mathbb{N}$, the WL test assigns the same color to $v$ and $v'$ at round $L$ if and only if there is an isomorphism between $\mathrm{Unr}_G^L(v)$ and $\mathrm{Unr}_{G'}^L(v')$ sending $v$ to $v'$.*

We will write $\mathrm{Unr}_G^L(v) \simeq \mathrm{Unr}_{G'}^L(v')$ to denote the existence of the isomorphism as in this observation. To prove Proposition C.1, we first rephrase Proposition 2.1 in terms of unravellings.

**Proposition C.4.** *Let $G$ and $G'$ be two graphs with nodes $v$ in $G$ and $v'$ in $G'$ such that $\mathrm{Unr}_G^L(v) \simeq \mathrm{Unr}_{G'}^L(v')$ for every $L \in \mathbb{N}$. Then for any AC-GNN $\mathcal{A}$, we have $\mathcal{A}(G, u) = \mathcal{A}(G', u')$.*

*Proof.* Follows directly from Proposition 2.1 and Observation C.3. $\qquad\square$

The crucial part of the proof of Proposition C.1 is the following non-trivial result, intuitively establishing that the fragment of unary FO formulas that only depend on the unravelling of a node is exactly the graded modal logic.

**Theorem C.5 (Otto, 2019).** *Let $\alpha$ be a unary FO formula. If $\alpha$ is not equivalent to a graded modal logic formula then there exist two graphs $G$, $G'$ and two nodes $v$ in $G$ and $u'$ in $G'$ such that $\mathrm{Unr}_G^L(v) \simeq \mathrm{Unr}_{G'}^L(v')$ for every $L \in \mathbb{N}$ and such that $u \models \alpha$ in $G$ but $u' \not\models \alpha$ in $G'$.*

*Proof.* This directly follows from the van Benthem & Rosen characterization obtained in (Otto, 2019, Theorem 2.2) for finite structures (graphs), by noticing that for the notion of graded bisimulation $\sim_\#$ introduced in this note, we have that $G, u \sim_\# G', u'$ if and only if we have that $\mathrm{Unr}_G^L(v) \simeq \mathrm{Unr}_{G'}^L(v')$ for every $L \in \mathbb{N}$. We point out here that the fact that the edge relation in $G$ is undirected in our setting (as opposed to $E$ being directed in (Otto, 2019)), and the fact that every node can only have one color in our setting (as opposed to being able to satisfy multiple "unary predicates" in (Otto, 2019)) are inessential, and that the proof of (Otto, 2019, Theorem 2.2) carries over to this setting. $\qquad\square$

We can now gather all of these to prove Proposition C.1.

*Proof of Proposition C.1.* Let $\alpha$ be a logical classifier (i.e., a unary FO formula) that is not equivalent to any graded modal logic formula. Assume for a contradiction that there exists an AC-GNN $\mathcal{A}_\alpha$ that captures $\alpha$. Since $\alpha$ is not equivalent to any graded modal logic formula, by Theorem C.5 there exist two graphs $G$, $G'$ and two nodes $v$ in $G$ and $u'$ in $G'$ such that $\mathrm{Unr}_G^L(v) \simeq \mathrm{Unr}_{G'}^L(v')$ for every $L \in \mathbb{N}$ and such that $(\star)$ $u \models \alpha$ in $G$ but $u' \not\models \alpha$ in $G'$. Since we have that $\mathrm{Unr}_G^L(v) \simeq \mathrm{Unr}_{G'}^L(v')$ for every $L \in \mathbb{N}$, by Proposition C.4 we should have that $\mathcal{A}_\alpha(G, u) = \mathcal{A}_\alpha(G', u')$. But this contradicts $(\star)$ and the fact that $\mathcal{A}_\alpha$ is supposed to capture $\alpha$. $\qquad\square$

# D    PROOF OF THEOREM 5.1

We first recall the theorem.

**Theorem 5.1.** *Each FOC$_2$ classifier can be captured by a simple homogeneous ACR-GNN.*

To prove the theorem, we will use a characterization of the unary FOC$_2$ formulas provided by (Lutz et al., 2001) that uses a specific modal logic. That logic is defined via what are called *modal parameters*. We adapt the definitions of (Lutz et al., 2001) to deal with simple undirected node-colored graphs.

**Definition D.1.** *A* modal parameter *is an expression built from the following grammar:*

$$S ::= \text{id} \mid e \mid S \cup S \mid S \cap S \mid \neg S.$$

*Given an undirected colored graph $G = (V, E)$ and a node $v$ of $G$, the interpretation of $S$ on $v$ is the set $\varepsilon_S(v) \subseteq V$ defined inductively as follows:*

– *if $S = \text{id}$ then $\varepsilon_S(v) := \{v\}$;*

– *if $S = e$ then $\varepsilon_S(v) := \{u \mid \{u, v\} \in E\}$;*

– *if $S = S_1 \cup S_2$ then $\varepsilon_S(v) := \varepsilon_{S_1}(v) \cup \varepsilon_{S_2}(v)$;*

– *if $S = S_1 \cap S_2$ then $\varepsilon_S(v) := \varepsilon_{S_1}(v) \cap \varepsilon_{S_2}(v)$;*

– *if $S = \neg S'$ then $\varepsilon_S(v) := V \setminus \varepsilon_S(v)$.*

*The modal logic $\mathcal{EMLC}$ consists of all the unary formulas that are built with the following grammar:*

$$\varphi ::= C \mid \varphi \wedge \varphi \mid \neg \varphi \mid \langle S \rangle^{\geq N} \varphi,$$

*where $C$ ranges over node colors, $S$ over modal parameters, and $N$ over $\mathbb{N}$. The semantics of the first four constructs is defined as expected, and for an undirected colored graph $G = (V, E)$ and node $v \in V$, we have $(G, v) \models \langle S \rangle^{\geq N} \varphi$ if and only if there exist at least $N$ nodes $u$ in $\varepsilon_S(v)$ such that $(G, u) \models \varphi$.*

**Example D.2.** *On an undirected graph $G = (V, E)$, the $\mathcal{EMLC}$ formula $\langle \neg e \rangle^{\geq 2}(\langle e \rangle^{\geq 3}\text{Green})$ holds on a node $v \in V$ if $v$ has at least two nonadjacent nodes $u$ (and since our graphs have no self-loops, $v$ could be $u$) such that $u$ has at least three green neighbors.*

The following theorem is essentially a reformulation of (Lutz et al., 2001, Theorem 1) to our context (Lutz et al. (2001) show this for FO$_2$ without counting quantifiers and for $\mathcal{EMLC}$ without counting, but an inspection of the proofs reveals that the result extends to counting quantifiers).

**Theorem D.3 (Lutz et al., 2001, Theorem 1).** *For every $\mathcal{EMLC}$ formula, there exists an equivalent FOC$_2$ unary formula. Conversely, for every unary FOC$_2$ formula, there exists an equivalent $\mathcal{EMLC}$ formula.*

In order to simplify the proof, we will use the following lemma.

**Lemma D.4.** *Let $\varphi$ be an $\mathcal{EMLC}$ formula. Then there exists an $\mathcal{EMLC}$ formula $\varphi'$ equivalent to $\varphi$ such that each modal parameter appearing in $\varphi'$ is one of the following:*

a) $\text{id}$, *thus representing the current node;*

b) $e$, *thus representing the neighbours of the current node;*

c) $\neg e \cap \neg \text{id}$, *thus representing the nodes distinct from the current node and that are not neighbours of the current node;*

d) $\text{id} \cup e$, *thus representing the current node and its neighbors;*

e) $\neg \text{id}$, *thus representing all the nodes distinct from the current node:*

f) $\neg e$, *thus representing the nodes that are not neighbours of the current node (note that this includes the current node);*

*g)* $e \cup \neg e$, *thus representing all the nodes;*

*h)* $e \cap \neg e$, *thus representing the emptyset.*

*Proof.* Let $v$ be a node in a graph $G$, and consider the following three disjoint sets of nodes:

1. the singleton set consisting of $v$ itself,

2. the set of neighbors of $v$,

3. the set of nodes that are not neighbors of $v$ and that are not $v$.

These sets can be expressed by modal parameters: the first is obtained by taking $S = \mathrm{id}$; the second is obtained by taking $S = e$; and the third is obtained by taking $S = \neg e \cap \neg \mathrm{id}$. It is straightforward to verify by induction on $S$ that, for any modal parameter $S$, if $\varepsilon_S(v)$ contains an element of one of the three sets, then it must contain all the elements of that set. But then, this implies that a modal parameter can only represent a (possibly empty) disjoint union of these three sets. Conversely, it is clear that any disjoint union over these three sets can be represented by a modal parameter. It is then routine to check that the 8 cases (a)–(h) are obtained as all the $2^3$ possible unions of these three sets (including the empty union, i.e., the emptyset). For instance, case (f) is the union of sets 1 and 3. □

*Proof of Theorem 5.1.* The proof is similar to that of Proposition 4.1. Let $\varphi$ be an $\mathcal{EMLC}$ formula equivalent to the targeted $\mathrm{FOC}_2$ unary formula that is of the form given by Lemma D.4, and let $\mathrm{sub}(\varphi) = (\varphi_1, \varphi_2, \ldots, \varphi_L)$ be an enumeration of the sub-formulas of $\varphi$ such that if $\varphi_k$ is a sub-formula of $\varphi_\ell$ then $k \leq \ell$. We will build a simple homogeneous ACR-GNN $\mathcal{A}_\varphi$ computing feature vectors $\boldsymbol{x}_v^{(i)}$ in $\mathbb{R}^L$ such that every component of those vectors represents a different formula in $\mathrm{sub}(\varphi)$. In addition, we will also make use of global feature vectors $\boldsymbol{x}_G^{(i)}$ in $\mathbb{R}^L$. The GNN $\mathcal{A}_\varphi$ will update the feature vector $\boldsymbol{x}_v^{(i)}$ of each node $v$ in a graph ensuring that component $\ell$ of $\boldsymbol{x}_v^{(i)}$ gets a value 1 if and only if the formula $\varphi_\ell$ is satisfied in node $v$ (and 0 otherwise). Similarly, $\boldsymbol{x}_G^{(i)}$ will be updated to make sure that every component represents the number of nodes in $G$ that satisfy the corresponding subformula. The readout and aggregate functions simply sum the input feature vectors. When $\varphi_\ell$ is of the form described by Cases 0–3 in the proof of Proposition 4.1, we define the $\ell$-th columns of the matrices $\boldsymbol{A}, \boldsymbol{C}$ and bias $\boldsymbol{b}$ as in that proof, and the $\ell$-th column of $\boldsymbol{R}$ (the matrix that multiplies the global readout feature vector) as the zero vector. We now explain how we define their $\ell$-th columns when $\varphi_\ell$ is of the form $\langle S \rangle^{\geq N} \varphi_k$, according to the 8 cases given by Lemma D.4:

*Case a.* if $\varphi_\ell = \langle \mathrm{id} \rangle^{\geq N} \varphi_k$, then $\boldsymbol{C}_{k\ell} = 1$ if $N = 1$ and 0 otherwise;

*Case b.* if $\varphi_\ell = \langle e \rangle^{\geq N} \varphi_k$, then $\boldsymbol{A}_{k\ell} = 1$ and $\boldsymbol{b}_\ell = -N + 1$;

*Case c.* if $\varphi_\ell = \langle \neg e \cap \neg \mathrm{id} \rangle^{\geq N} \varphi_k$, then $\boldsymbol{R}_{k\ell} = 1$ and $\boldsymbol{C}_{k\ell} = \boldsymbol{A}_{k\ell} = -1$ and $\boldsymbol{b}_\ell = -N + 1$;

*Case d.* if $\varphi_\ell = \langle \mathrm{id} \cup e \rangle^{\geq N} \varphi_k$, then $\boldsymbol{C}_{k\ell} = 1$ and $\boldsymbol{A}_{k\ell} = 1$ and $\boldsymbol{b}_\ell = -N + 1$;

*Case e.* if $\varphi_\ell = \langle \neg \mathrm{id} \rangle^{\geq N} \varphi_k$, then $\boldsymbol{R}_{k\ell} = 1$ and $\boldsymbol{C}_{k\ell} = -1$ and $\boldsymbol{b}_\ell = -N + 1$;

*Case f.* if $\varphi_\ell = \langle \neg e \rangle^{\geq N} \varphi_k$, then $\boldsymbol{R}_{k\ell} = 1$ and $\boldsymbol{A}_{k\ell} = -1$ and $\boldsymbol{b}_\ell = -N + 1$;

*Case g.* if $\varphi_\ell = \langle e \cup \neg e \rangle^{\geq N} \varphi_k$, then $\boldsymbol{R}_{k\ell} = 1$ and $\boldsymbol{b}_\ell = -N + 1$;

*Case h.* if $\varphi_\ell = \langle e \cap \neg e \rangle^{\geq N} \varphi_k$, then all relevant values are 0;

and all other values in the $\ell$-th columns of $\boldsymbol{A}, \boldsymbol{C}, \boldsymbol{R}$, and $\boldsymbol{b}$ are 0. The proof then goes along the same lines as the proof of Proposition 4.1. □

## E    PROOF OF THEOREM 5.2

We first recall the theorem.

**Theorem 5.2.** *Each FOC$_2$ classifier is captured by an AC-FR-GNN.*

In the following proof we will use the machinery introduced in Appendices C and D. We will also make use of a particular AC-GNN with $L$ layers, which we call $\mathcal{A}^L_{\mathrm{primes}}$, that maps every node $v$ in a graph $G$ to a natural number representing the complete unravelling of $v$ of depth $L$ in $G$ (note that we do not claim that this AC-GNN can be realized in practice, this construction is mostly for theoretical purposes). Let primes : $\mathbb{N} \to \mathbb{N}$ be the function such that primes$(i)$ is the $i$-th prime number indexed from 0. For instance, we have that primes$(0) = 2$, primes$(1) = 3$, etc. Now consider the function f$(\cdot, \cdot)$ that has as input a pair $(c, X)$ where $c \in \mathbb{N}$ and $X$ is a multiset of numbers in $\mathbb{N}$, and produces a number in $\mathbb{N}$ as output, defined as follows

$$\mathrm{f}(c, \{\!\{x_1, x_2, \ldots, x_k\}\!\}) = 2^c \times \prod_{i=1}^{k} \mathrm{primes}(x_i + 1).$$

It is not difficult to prove that, as defined above, f$(\cdot, \cdot)$ is an injective function. Thus using the results by Xu et al. (2019) (see the proof of their Theorem 3) we know that f can be used to implement the combine and aggregate operators of an AC-GNN such that for every graph $G$, after $L$ layers, the color (natural number) assigned to every node in $G$ has a one to one correspondence with the color assigned to that node in the $L$-th iteration of the WL test over $G$. We call this AC-GNN $\mathcal{A}^L_{\mathrm{primes}}$.

**Observation E.1.** *We note that Xu et al. (2019) also constructed an injective function that has $(c, X)$ as inputs where $c \in \mathbb{N}$ and $X$ is a multiset of elements in $\mathbb{N}$ (see their Lemma 5 and Corollary 6). Nevertheless we cannot directly use that construction as it assumes the existence of a fixed $N$ such that the size of all multisets are bounded by $N$. This would put also a bound of $N$ on the maximum number of neighbors in the input graphs. Thus we developed a new function (using an encoding based on prime numbers) to be able to deal with general graphs of unbounded degree.*

*Proof of Theorem 5.2.* Let $\alpha$ be an FOC$_2$ unary formula, and let $\varphi$ be an equivalent $\mathcal{EMLC}$ formula that uses only modal parameters of the form given by Lemma D.4. We construct an ACR-FR-GNN $\mathcal{A}_\varphi$ capturing $\varphi$ and hence $\alpha$.

Let $L$ be the quantifier depth of $\varphi$ (i.e., the deepest nesting of $\langle S \rangle^{\geq N}$ quantifiers). For a subformula $\varphi'$ of $\varphi$, we also define the *nesting depth* $\mathrm{nd}_\varphi(\varphi')$ *of* $\varphi'$ *in* $\varphi$ to be the number of modal parameters under which $\varphi'$ is in $\varphi$. The first $L - 1$ layers of $\mathcal{A}_\varphi$ are the same as those of $\mathcal{A}^{L-1}_{\mathrm{primes}}$, which do not use readouts. With Observation C.3 at hand and using the fact that the inverses of the aggregation and combination functions of $\mathcal{A}^{L-1}_{\mathrm{primes}}$ are computable, this ensures that, after $L - 1$ layers, for any graph $G$ and node $v$ in $G$, we can compute from $\mathcal{A}^{L-1}_{\mathrm{primes}}(G, v)$ the unravelling $\mathrm{Unr}^{L-1}_G(v)$. Thus, we can assume without loss of generality (by modifying the last combination function for instance), that after $L - 1$ layers $\mathcal{A}_\varphi$ computes $\mathrm{Unr}^{L-1}_G(v)$ in every node $v$ of $G$. We then use a readout whose output is a natural number representing the multiset $\{\!\{\mathrm{Unr}^{L-1}_G(v) \mid v \text{ node in } G\}\!\}$; for instance, we can encode this multiset using the same technique that we use for $\mathcal{A}_{\mathrm{primes}}$. Again, since this technique uses functions with computable inverses, we can assume without loss of generality that the output of this readout is actually the multiset $\{\!\{\mathrm{Unr}^{L-1}_G(v) \mid v \text{ node in } G\}\!\}$. Finally, we use a final combination function $\mathrm{COM}^{(L)}$, that uses only the feature of the current node and the output of the readout—that is, the final feature of a node $v$ is $\mathrm{COM}^{(L)}(\mathrm{Unr}^{L-1}_G(v), \{\!\{\mathrm{Unr}^{L-1}_G(u) \mid u \text{ node in } G\}\!\})$.

We now explain how we define $\mathrm{COM}^{(L)}$. By induction on the structure of $\varphi$, for every subformula $\varphi'$ of $\varphi$, we do the following: for every node $v$ in $G$ and every node $u$ in $\mathrm{Unr}^{L-1}_G(v)$ that is at depth (i.e., the distance from $v$) at most $\mathrm{nd}_\varphi(\varphi')$ in the tree $\mathrm{Unr}^{L-1}_G(v)$, we will label $u$ by either $\varphi'$ or by $\neg\varphi'$. We do so to ensure that $(\star)$ for every node $v$ in $G$ and every node $u = (v, u_1, \ldots, u_i)$ in $\mathrm{Unr}^{L-1}_G(v)$, we label $u$ by $\varphi'$ if and only if $(G, u_i) \models \varphi'$. We explain our labeling process by induction on the structure of $\varphi$, and one can easily check in each case that $(\star)$ will hold by induction. Let $v$ be a node in $G$ and $u$ be a node in $\mathrm{Unr}^{L-1}_G(v)$ that is at depth at most $\mathrm{nd}_\varphi(\varphi')$ in the unravelling.

*Case 1.* If $\varphi'$ is a color Col, we label $u$ by $\varphi'$ if $u$ is of that color, and by $\neg\varphi'$ otherwise.

*Case 2.* If $\varphi'$ is $\varphi_1 \wedge \varphi_2$, then observe that we have $\mathrm{nd}_\varphi(\varphi') = \mathrm{nd}_\varphi(\varphi_1) = \mathrm{nd}_\varphi(\varphi_2)$, so that $u$ is at depth at most both $\mathrm{nd}_\varphi(\varphi_1)$ and $\mathrm{nd}_\varphi(\varphi_2)$ in the unravelling $\mathrm{Unr}^{L-1}(v)$. Thus, we know that we have already labeled $u$ by either $\varphi_1$ or $\neg\varphi_1$, and also by either $\varphi_2$ or $\neg\varphi_2$. We then label $u$ by $\varphi'$ if $u$ is already labeled by $\varphi_1$ and $\varphi_2$, and we label it by $\neg\varphi'$ otherwise.

*Case 3.* The case when $\varphi'$ is a negation is similar.

*Case 4.* If $\varphi'$ is $\langle S \rangle^{\geq N} \varphi''$, then we only explain the case when the modal parameter $S$ is $\neg e \wedge \neg \mathrm{id}$, as the other cases work similarly. First, observe that for every node $v'$ in $G$, we have labeled the root of $\mathrm{Unr}_G^{L-1}(v')$ by either $\varphi''$ or by $\neg \varphi''$: this is because the root of $\mathrm{Unr}_G^{L-1}(v')$ is always at depth $0 \leq \mathrm{nd}_\varphi(\varphi'')$ in $\mathrm{Unr}_G^{L-1}(v')$. Let $m$ be the number of nodes $u' \in G$ such that we have labeled the root of $\mathrm{Unr}_G^{L-1}(v')$ by $\varphi''$. Next, note that for every children $u'$ of $u$ in $\mathrm{Unr}_G^{L-1}(v)$, we have that $u'$ is at depth at most $\mathrm{nd}_\varphi(\varphi'')$ in $\mathrm{Unr}_G^{L-1}(v)$, so that we have already labeled $u'$ by either $\varphi''$ or $\neg \varphi''$. Let $n$ be the number of children of $u$ (in $\mathrm{Unr}_G^{L-1}(v)$) that we have labeled by $\varphi''$. Then we label $u$ by $\varphi'$ if $m - n \geq N$, and by $\neg \varphi'$ otherwise.

We then simply define $\mathrm{COM}^{(L)}(\mathrm{Unr}_G^{L-1}(v), \{\!\{\mathrm{Unr}_G^{L-1}(u) \mid u \text{ node in } G\}\!\})$ to be 1 if the root of $\mathrm{Unr}_G^{L-1}(v)$ is labeled with $\varphi$, and 0 otherwise, which concludes the proof. $\qquad\square$

## F    DETAILS ON THE EXPERIMENTAL SETTING AND RESULTS

All our code and data can be accessed online at `https://github.com/juanpablos/GNN-logic`

In all our experiments we tested different aggregate, combine and readout functions. For aggregate and readout we only consider the sum, average, and max functions. For the combine function we consider the following variants:

- $\mathrm{COM}_1(\boldsymbol{x}, \boldsymbol{y}, \boldsymbol{z}) = f(\boldsymbol{x}\boldsymbol{A} + \boldsymbol{y}\boldsymbol{B} + \boldsymbol{z}\boldsymbol{C} + \boldsymbol{b})$,
- $\mathrm{COM}_2(\boldsymbol{x}, \boldsymbol{y}, \boldsymbol{z}) = f(\mathrm{MLP}_1(\boldsymbol{x}) + \mathrm{MLP}_2(\boldsymbol{y}) + \mathrm{MLP}_3(\boldsymbol{z}) + \boldsymbol{b})$,
- $\mathrm{COM}_3(\boldsymbol{x}, \boldsymbol{y}, \boldsymbol{z}) = \mathrm{MLP}(\boldsymbol{x} + \boldsymbol{y} + \boldsymbol{z} + \boldsymbol{b})$,
- $\mathrm{COM}_4(\boldsymbol{x}, \boldsymbol{y}, \boldsymbol{z}) = \mathrm{MLP}(\boldsymbol{x}\boldsymbol{A} + \boldsymbol{y}\boldsymbol{B} + \boldsymbol{z}\boldsymbol{C} + \boldsymbol{b})$.

The above definitions are for ACR-GNNs. For AC-GNNs we consider similar variants but without the $\boldsymbol{z}$ input. We also used batch normalization in between every GNN and MLP layer. We did not use any regularization. When processing synthetic data we use a hidden size of 64 and trained with a batch-size of 128, and the Adam optimizer with PyTorch default parameters for 50 epochs. We did not do any hyperparameter search besides changing the aggregation, combination, and readout functions. For the activation functions we always used relu. We observed a consistent pattern in which sum aggregator and readout produced better results compared with the others. This is in line with our constructions in Proposition 4.1 and Theorem 5.1. The choice of the combination function did not produce a significant difference in the performance.

DATA FOR THE EXPERIMENT WITH CLASSIFIER $\alpha(x) := \mathrm{RED}(x) \wedge \exists y \, \mathrm{BLUE}(y)$

For training and testing we constructed three sets of graphs: (a) Train set containing 5k graphs with nodes between 50 and 100, (b) Test set, same size, containing 500 graphs with the same number of nodes as in the train set (between 50 and 100 nodes), and (c) Test set, bigger size, containing 500 graphs with nodes between 100 and 200. All graphs contain up to 5 different colors. To force the models to try to learn the formula, in every set (train and test) we consider 50% of graphs not containing any *blue* node, and 50% containing at least one *blue* node. The number of *blue* nodes in every graph is fixed to a small number (typically less than 5 nodes). Moreover, to ensure that there is a significant number of nodes satisfying the formula, we force graphs to contain at least 1/4 of its nodes colored with *red*. The colors of all the other nodes are distributed randomly. With all these restrictions, every dataset that we created had at least a 18% of nodes satisfying the property. We consider two classes of graphs: **line graphs** and **Erdös-Renyi graphs**.

**Line graphs**    these are connected graphs in which every node in the graph has degree 2 except for two nodes (the extreme nodes) that have degree 1. To mimic the impossibility proof in Proposition 3.3 we put the *blue* nodes in one of the "sides" of the line, and the *red* nodes in the other "side". More specifically, consider the line graph with $N$ nodes $v_1, \ldots, v_N$ such that $v_i$ is connected with $v_{i+1}$. Then, we ensure that every *blue* node appears in one of $v_1, \ldots, v_{\frac{N}{2}}$ and every *red* node appears in one of $v_{\frac{N}{2}+1}, \ldots, v_N$.

|  | # Graphs | Avg. # Nodes | Avg. # Edges | Avg. # Positive |
|---|---|---|---|---|
| Line train | 5,000 | 75 | 74 | 18 |
| Line test | 500 | 75 | 74 | 18 |
| Line test bigger | 500 | 148 | 147 | 36 |
| Erdös-Renyi train | 5,000 | 75 | 115 | 18 |
| Erdös-Renyi test | 500 | 75 | 115 | 18 |
| Erdös-Renyi test bigger | 500 | 148 | 226 | 36 |

Table 3: Synthetic data for the experiment with classifier $\alpha(x) := \text{Red}(x) \wedge \exists y \, \text{Blue}(y)$

|  | Erdös-Renyi + 20% | | | Erdös-Renyi + 50% | | | Erdös-Renyi + 100% | | |
|---|---|---|---|---|---|---|---|---|---|
|  | Train Acc. | Test Acc. | | Train Acc. | Test Acc. | | Train Acc. | Test Acc. | |
|  |  | same-size | bigger |  | same-size | bigger |  | same-size | bigger |
| AC-2 | 0.810 | 0.807 | 0.778 | 0.829 | 0.835 | 0.791 | 0.861 | 0.864 | 0.817 |
| AC-5 | 0.940 | 0.937 | 0.901 | 0.975 | 0.971 | 0.958 | 0.994 | 0.994 | 0.993 |
| AC-7 | 0.963 | 0.961 | 0.946 | 0.983 | 0.978 | 0.981 | 0.995 | 0.995 | 0.995 |
| GIN-2 | 0.797 | 0.795 | 0.771 | 0.813 | 0.818 | 0.784 | 0.838 | 0.840 | 0.803 |
| GIN-5 | 0.838 | 0.836 | 0.819 | 0.846 | 0.847 | 0.833 | 0.841 | 0.844 | 0.838 |
| GIN-7 | 0.838 | 0.840 | 0.803 | 0.841 | 0.844 | 0.838 | 0.784 | 0.788 | 0.773 |
| ACR-1 | 1.000 | 1.000 | 1.000 | 1.000 | 1.000 | 1.000 | 1.000 | 1.000 | 1.000 |

Table 4: Detailed results for Erdös-Renyi synthetic graphs with different connectivities

**Erdös-Renyi graphs**   These are random graphs in which one specifies the number $N$ of nodes and the number $M$ of edges. For this experiment we consider as extreme cases the case in which graphs contain the same number of nodes and edges and graphs in which the number of edges is twice the number of nodes.

Some statistics of the datasets are shown in Table 3.

EXPERIMENTS FOR DENSE ERDÖS-RENYI GRAPHS

We also took a closer look at the performance for different connectivities of random graphs (Table 4). We define the set "Erdös-Renyi + $k\%$" as a set of graphs in which the number of edges is $k\%$ larger than the number of nodes. For example, "Erdös-Renyi + 100%" contains random graphs in which the number of egdes doubles the number of nodes. We see a consistent improvement in the performance of AC-GNNs and GINs when we train and test them with more dense graphs and more layers (Table 4).

DATA FOR THE EXPERIMENT WITH CLASSIFIER $\alpha_i(x)$ IN EQUATION (6)

For this case we only consider dense Erdös-Renyi synthetic graphs. For the train set we consider graphs with nodes varying from 40 to 50 nodes and edges from 280 to 350 and similarly for the first test set. For the bigger test set, we consider graphs with nodes from 51 to 60 with edges ranging from 360 and 480. For labeling we consider the following formulas (starting from $\alpha_0(x) := \text{Blue}(x)$):

$$\begin{aligned}
\alpha_1(x) &:= \exists^{[8,10]} y \big( \alpha_0(y) \wedge \neg E(x,y) \big), \\
\alpha_2(x) &:= \exists^{[10,20]} y \big( \alpha_1(y) \wedge \neg E(x,y) \big), \\
\alpha_3(x) &:= \exists^{[10,30]} y \big( \alpha_2(y) \wedge \neg E(x,y) \big).
\end{aligned}$$

The choices of the intervals for every classifier were for the pourpose of having approximately half of the nodes in the random graphs marked as true. Statistics of the datasets are shown in Table 5.

|              | # Graphs | Avg. # Nodes | Avg. # Edges | Pos. $\alpha_1$ | Pos. $\alpha_2$ | Pos. $\alpha_3$ |
|--------------|----------|--------------|--------------|--------|--------|--------|
| Train        | 5,000    | 45           | 315          | 47%    | 63%    | 57%    |
| Test         | 500      | 45           | 315          | 47%    | 64%    | 56%    |
| Test bigger  | 500      | 56           | 420          | 49%    | 40%    | 23%    |

Table 5: Synthetic data for the experiment with classifier $\alpha_i(x)$ in Equation (6)

|        | F1 Test          |
|--------|------------------|
| AC-2   | $97.2 \pm 0.3$   |
| AC-3   | $97.5 \pm 0.3$   |
| AC-4   | $97.5 \pm 0.2$   |
| ACR-2  | $93.5 \pm 0.3$   |
| ACR-3  | $94.2 \pm 1.2$   |
| ACR-4  | $95.4 \pm 0.9$   |

Table 6: Performance of AC-GNN and ACR-GNN in the PPI benchmark

PPI EXPERIMENTS

We consider the standard train/validation/test split for this benchmarck (Fey & Lenssen, 2019). We use a hidden size of 256 and the Adam optimizer for 500 epochs with early stopping when the validation set did not improve for 20 epochs. We did not do any hyperparameter search besides changing the aggregation, combination, and readout functions. As opposed to the synthetic case, in this case we observed a better performance when the average or the max functions are used for aggregation. Table 6 shows the best results for different layers (average of 10 runs). As we can see, ACR-GNNs do not imply an improvement over AC-GNNs for this benchmark.

