# OpenReview forum: "The Logical Expressiveness of Graph Neural Networks"
_ICLR.cc/2020/Conference — Accept (Spotlight)_

### Official Review · AnonReviewer1 · 2019-10-23
**Official Blind Review #1**

**Rating:** 8

**Review:**

The paper utilizes recent insights into the relationship between the Weisfeiler-Lehman (WL) test for checking graph isomorphism and Graph Neural Networks (GNNs) in order to characterize the class of node classifiers that can be captured by a specific GNN architecture, called aggregate-combine GNN (AC-GNN).

The primary contribution of this work is the identification of the logical classifiers that can be represented within an AC-GNN, a fragment of first order logic called graded modal logic, as well as an extention of AC-GNNs with the ability to capture a strictly more expressive fragment of first order logic, called ACR-GNN. Both of these results are supported by formal proofs and derivations, adding not only theoretical value to the presented work, but also intuitive insights behind the reasons of the AC-GNN limitations. In addition, the presented experiments demonstrate the practical implications of the results, with the exception of some datasets where no significance difference in performance between AC-GNNs and ACR-GNNs was found.
I believe that the contributions of this paper are significant. On the one hand, studying theoretical properties of GNNs facilitates their transparency and highlights the range of applications they can be used in. On the other hand, although the GNN variant introduced by the author is a special case of an existing class of GNNs, the motivation behind its introduction is different and follows naturally from the discussion within the paper.

The fact that no actual difference in performance between AC-GNNs and ACR-GNNs was noticed in the only non-synthetic dataset used in the experiment should prompt the author to run experiments with more real life datasets, in order to empirically verify the results, but this is a minor point.

**Experience Assessment:**

I have published one or two papers in this area.

**Review Assessment: Checking Correctness Of Derivations And Theory:**

I did not assess the derivations or theory.

**Review Assessment: Checking Correctness Of Experiments:**

I did not assess the experiments.

**Review Assessment: Thoroughness In Paper Reading:**

I read the paper at least twice and used my best judgement in assessing the paper.

---

> ### Author Response · Authors · 2019-11-12
> **Answer to reviewer #1**
>
> Thank you for your review. You correctly point out that more empirical results on real life data are needed and this is part of our current and future work. We just add that there is some work showing that current benchmarks might not be very helpful when differentiating specific changes for GNNs [1]. It is even common practice to only keep and train over the biggest connected component of the graph, and discard the rest, so our ACR-GNN might not benefit from those situations. Recent work has also started studying that current real life datasets might not be adequate for some tasks [2]. Moreover, most common benchmarks do not encode global constraints in the graph features nor labels, and thus it might not be easy to find a good benchmark to show for our global approach. We comment on this only to emphasize that finding a good benchmark for our approach has not been an easy task and that is why we focused more on synthetic data in this submission.
>
> [1] Chen, Ting, Song Bian, Yizhou Sun. "Are Powerful Graph Neural Nets Necessary? A Dissection on Graph Classification." https://arxiv.org/abs/1905.04579
> [2] Ivanov, Sergei, Sergei Sviridov, and Evgeny Burnaev. "Understanding Isomorphism Bias in Graph Data Sets." https://arxiv.org/abs/1910.12091

---

### Official Review · AnonReviewer2 · 2019-10-23
**Official Blind Review #2**

**Rating:** 8

**Review:**

The paper elaborates on the expressivity of graph neural networks (GNNs). More precisely, the authors show that expressivity of AC-GNNs (aggregate and combine) can only express logical classifiers that can be expressed in graded modal logic. By adding readouts, ACR-GNNs (aggregate, combine and readout) can capture FOC2 which is logical classifiers expressed with 2 variables and counting quantifiers. The second theorem leaves open the question of whether ACR-GNNs can capture logical classifiers beyond FOC2.

The paper is written nicely, its easy on the eyes, and delegates the proofs to the appendix. I was a bit surprised by the lack of a discussion connecting the choice of the aggregate and combine operations to the representation power of GNNs. One has to delve deep into the proofs to find out if the choice of these operations affects expressivity.

**Experience Assessment:**

I have read many papers in this area.

**Review Assessment: Checking Correctness Of Derivations And Theory:**

I assessed the sensibility of the derivations and theory.

**Review Assessment: Checking Correctness Of Experiments:**

I assessed the sensibility of the experiments.

**Review Assessment: Thoroughness In Paper Reading:**

I read the paper at least twice and used my best judgement in assessing the paper.

---

> ### Author Response · Authors · 2019-11-12
> **Answer to reviewer #2**
>
> Thank you very much for your comments. In the final version of the paper, we plan to add in the body of the paper more discussion on the aggregate and combine operation used in the proofs and how they affect the expressiveness of GNNs.

---

### Official Review · AnonReviewer3 · 2019-11-05
**Official Blind Review #3**

**Rating:** 8

**Review:**

This paper establishes novel theoretical connections between Boolean node classifiers on aggregate-combine Graph Neural Networks (AC-GNNs) and first-order predicate logic (FOC2). It shows that current boolean node classifiers on AC-GNNs can only represent a subset of FOC2 but that a simple extension taking into global information can generalise AC-GNNs to the full FOC2.

The style of the manuscript is mixed: the abstract and introduction are quite dense for non-experts; a motivating real-world example could help here. The other sections, on the other hand, are quite good to follow and most concepts have a good high-level introduction as well as a little example to follow the arguments.

The theoretical connections connection between GNNs and first-order logic strike me as interesting. I did not, however, understand the results reported in table 2: ACR reaches optimal performance only on alpha_1 but not on alpha_2 and alpha_3. Does this happen because the latter two expressions are not part of FOC2?

---
I increased my rating due to the author rebuttal.

**Experience Assessment:**

I do not know much about this area.

**Review Assessment: Checking Correctness Of Derivations And Theory:**

I did not assess the derivations or theory.

**Review Assessment: Checking Correctness Of Experiments:**

I assessed the sensibility of the experiments.

**Review Assessment: Thoroughness In Paper Reading:**

I made a quick assessment of this paper.

---

> ### Author Response · Authors · 2019-11-12
> **Answer to reviewer #3**
>
> Thanks a lot for your positive comments. We will do our best to improve the readability of the introduction, and to try to fit an example. In any case we will delay this change for an eventual final version of the paper as it might require some more general changes to the flow of the intro. But thank you very much for pointing this out.
>
> Regarding your question, all \alpha_i do belong to FOC2. In fact, it is not difficult to show that there exists an ACR-i GNN that captures alpha_i for each i, but none of ACR-j GNNs with j < i can do this. Nonetheless, the existence of such a GNN does not imply the ability to learn this GNN from examples. So, besides other things, our experiments demonstrate that, for i = 2 and i = 3, it is possible to learn an ACR-i GNN very close (~80-90%) to alpha_i, in contrast to AC-GNNs that achieve a lower performance (60%) even if 10 layers are allowed; The question whether and how it is possible to learn an ACR-i GNN exactly equivalent to alpha_i is still open. We will add a discussion about this into the final version of the paper.

---

### Public Comment · ~Ryoma_Sato1 · 2019-11-08
**A related work on the connection between GNNs and modal logic**

Hi, the analysis of the connection between FOC2 and GNNs is interesting, and I really enjoyed reading it. [Sato+ 2019] pointed out that the GNN classes are equivalent to computational models of local algorithms. Especially, it gave a theoretical consideration about the limit of the ability of GNNs in terms of approximation ratios by pointing out the connections between GNNs and local algorithms. Since each computational model of local algorithms is equivalent to a modal logic class such as graded modal logic [Hella+ 2012], the contribution of this paper is closely related to [Sato+ 2019], and it is good for this paper to refer to them.

[1] Ryoma Sato, Makoto Yamada, Hisashi Kashima. Approximation Ratios of Graph Neural Networks for Combinatorial Problems. NeurIPS 2019. https://arxiv.org/abs/1905.10261

[2] Lauri Hella, Matti Järvisalo, Antti Kuusisto, Juhana Laurinharju, Tuomo Lempiäinen, Kerkko Luosto, Jukka Suomela, Jonni Virtema. Weak Models of Distributed Computing, with Connections to Modal Logic. PODC 2012. https://arxiv.org/abs/1205.2051

Thank you for your attention.

---

> ### Author Response · Authors · 2019-11-12
> **Answer to related work**
>
> Thank you for these relevant pointers. Indeed, it seems likely that a result similar to our Proposition 4.1 (saying that each graded modal logic formula can be captured by an AC-GNN) could be derived by the combination of these two papers: by Theorem 2, item (f) of [Hella et al., 2012], the GML formula \phi would be captured by an “MB” distributed algorithm A_\phi, which in turn, by the MB version of  Lemma 13 of [Sato et al., 2019], would be captured by an AC-GNNt. However, an important part of our Proposition 4.1 is that we use AC-GNNs that are “simple”, in the sense defined in our paper (aggregation is a matrix multiplication of the sum of the feature vectors of the neighbors, and we use a (truncated) ReLU); this in contrast does not seem to be what we would obtain by combining the results of [Sato et al., 2019] and [Hella et al., 2012]. In addition, we observe the following differences:
> - [Sato et al., 2019] considers graphs of bounded degree, while our graphs can have unbounded degree;
> - the forward direction of our Theorem 4.2 (i.e., if an FO formula is not equivalent to a GML formula then no AC-GNN can capture it) does not follow from these two papers, since it goes outside of modal logics;
> - more generally, [Sato et al., 2019] and [Hella et al., 2012] consider only “local” variants of GNNs and of distributed algorithms, i.e., (1) no internal global readouts in the GNNs; (2) in the distributed algorithms the messages are always transmitted only to the neighbors; and (3) the logics considered are all modal i.e., local; therefore, all our results on ACR-GNNs and FO and FO_2 do not seem to follow from these papers.
>
> We will point to these works in the final version.

---

### Decision · Program_Chairs · 2019-12-19

**Decision:**

Accept (Spotlight)

**Comment:**

The paper focuses on characterizing the expressiveness of graph neural networks. The reviewers were satisfied that the authors answered their questions suffciiently and uniformly agree that this is a strong paper that should be accepted.